# Compact Magnetic Force Microscope (MFM) System in a 12 T Cryogen-Free Superconducting Magnet

**DOI:** 10.3390/mi13111922

**Published:** 2022-11-07

**Authors:** Asim Abas, Tao Geng, Wenjie Meng, Jihao Wang, Qiyuan Feng, Jing Zhang, Ze Wang, Yubin Hou, Qingyou Lu

**Affiliations:** 1High Magnetic Field Laboratory, Hefei Institutes of Physical Science, Chinese Academy of Sciences, Hefei 230031, China; 2Department of Hefei National Research Center for Physical Sciences at the Microscale, University of Science and Technology of China, Hefei 230026, China; 3Anhui Laboratory of Advanced Photon Science and Technology, University of Science and Technology of China, Hefei 230026, China; 4Anhui Province Key Laboratory of Condensed Matter Physics at Extreme Conditions, High Magnetic Field Laboratory of Anhui, Hefei 230031, China; 5Hefei Science Center, Chinese Academy of Sciences, Hefei 230031, China

**Keywords:** magnetic force microscope, high magnetic field, piezoelectric tube, cryogen-free superconducting

## Abstract

Magnetic Force Microscopy (MFM) is among the best techniques for examining and assessing local magnetic characteristics in surface structures at scales and sizes. It may be viewed as a unique way to operate atomic force microscopy with a ferromagnetic tip. The enhancement of magnetic signal resolution, the utilization of external fields during measurement, and quantitative data analysis are now the main areas of MFM development. We describe a new structure of MFM design based on a cryogen-free superconducting magnet. The piezoelectric tube (PZT) was implemented with a tip-sample coarse approach called SpiderDrive. The technique uses a magnetic tip on the free end of a piezo-resistive cantilever which oscillates at its resonant frequency. We obtained a high-quality image structure of the magnetic domain of commercial videotape under extreme conditions at 5 K, and a high magnetic field up to 11 T. When such a magnetic field was gradually increased, the domain structure of the videotape did not change much, allowing us to maintain the images in the specific regions to exhibit the performance. In addition, it enabled us to locate the sample region in the order of several hundred nanometers. This system has an extensive range of applications in the exploration of anisotropic magnetic phenomena in topological materials and superconductors.

## 1. Introduction

For decades, the most progressive idea in Magnetic Force Microscopy (MFM) research seems to have been that microscopic magnetic imaging has some of the most effective tools for studying the dynamic characteristics of magnetic domains and magnetic phase transitions of magnetic properties under magnetic field instrumentation. MFMs are mainly used in the research of magnetic storage materials. Many magnetically anisotropic materials strongly connect their magnetic characteristics [1,2,3]. MFMs operating under harsh conditions are becoming widespread due to their various significance in the magnetic recording industry [4,5,6,7]. Several macroscopic observations of magnetic materials, including resistivity, susceptibility, thermal conductivity, or specific heat, are constructed by placing a sample holder in a magnetic field [8,9,10,11]. Although numerous samples exhibited transitions at both low temperatures in addition to magnetic fields, MFMs under low temperatures and strong magnetic fields are also still rare, which is essential [12,13]. Many instruments with different principles and functions are being developed. Despite having obtained excellent results using self-made microscopes in our research group, there are still many problems in practical application. Specifically, there are the following problems: (1) poor walking of the stepper motor because of the metal-metal walking mode. This is especially serious in low-temperature experiments, which reduce working efficiency. (2) The imaging quality still requires improvement, and the resolution of samples with finer magnetic structures is still insufficient. (3) The temperature drift is severe, resulting in the need to constantly adjust the position and the vertical distance between the sample and the tip during the measurement process. This leads to a cumbersome measurement process and affects image quality. (4) The position between the sample and the tip is relatively fixed, and the sample area can only be searched by relying on the scanning range of the tube itself. This makes the measurement process very time-consuming, and much time is spent pointlessly looking for the sample area. Constructing an MFM with a strong magnetic field and low temperature is challenging because of the complicated and demanding criteria for low drift, high precision, rigidity, and compactness. A serious problem occurs under this device: even though a cryostat with a liquid helium bath exhibits less vibration, it still needs a considerable amount of liquid helium. Because helium is a rare and non-renewable resource, its price may rise soon; there is a worldwide helium scarcity, and the situation for fundamental research is predicted to worsen [14,15]. Aside from the high liquid helium demand, the unique design of such a cryostat, which would be required to keep a lengthy holding duration and demands an intensive labor investment to manage the device, represents a substantial expense for researchers. This solution is widely known: it uses a cryogen-free magnet system; it does not use liquid helium. A pulse-tube cryocooler is used to cool these superconducting coils, and a Gifford-McMahon (GM) cryocooler is simple to operate. The continuously rising number of applications for cryogen-free magnet systems demonstrates that they are the future trend. In the present work, we have designed and constructed a high-resolution MFM that can measure the device under extreme conditions. The whole structure design can solve the previously mentioned problem. We optimized the process in terms of the equipment. Different materials were tested, and the ceramic element was chosen as the core part of a stepper motor to make the low-temperature function operate smoothly. At the same time, we adopted a compact lens structure with strong overall rigidity, reducing the temperature drift and significantly enhancing the imaging quality.

## 2. Materials and Methods

The cryogen-free superconducting magnets 12 T were used by the MFM system. The probe used in this MFM is a piezo-resistive cantilever with self-sensing capability (PRSA-L300-F50-STD from SCL-sensor, fabrication Gmbh). Using our homemade active Wheatstone bridge, the second stage is matched with a 500-time amplifier to detect the resistance change of the piezo-resistive cantilever. The cantilever beam’s tip has to be covered with a magnetic coating in order to detect magnetic force. MFM enables the imaging of magnetic structures, which are often only tens of nanometers in size. Various methods have been used to manufacture ultra-high precision MFM in order to achieve a higher resolution level, which is essentially determined by the size of the magnetic tip apex. In order to fabricate a high-resolution cantilever tip, several of the most promising methods depend on ion or scanning electron beam technology. A completely different strategy employs a tiny multilayered (ML) coating, which results in the presence of a nanoscale magnetic loss area at the probe’s apex. This coating is made up of two layers of magnetic material, sandwiched by a non-magnetic interlayer [16]. Therefore, we used electron beam evaporation (using the ZZS-630 electron beam evaporation machine from Xing-nan technology company in Shanghai China). In the process of evaporation, it is necessary to keep an angle between the evaporation source and the cantilever tip, so that only the tip of the needle is coated with the metal film. Special attention is required to protect the circuit part to sequentially coat the tip with three layers of the film: first, a 5 nm titanium film is deposited as the buffer layer; then, a 50 nm cobalt film is laid down; and finally, a 5 nm gold film is applied to cover it all as a protective layer [17]. Figure 1a–d shows the thickness of the SEM image of the cantilever tip coated with different magnifications of 100 nm, 200 nm, 300 nm, and 500 nm respectively. The ferromagnetic film coated on tip of the piezoresistive cantilever for magnetic force microscope for application. Only this uncompensated region, where the magnetic material is “effectively localized” close to the probe’s apex, participates actively in the interaction between the MFM tip and the sample [18]. We utilized the R9 controller (RHK Technology) for the MFM imaging experiment. The R9 controller was used for MFM scanning control and signal processing. MFM images were collected in a constant height mode. We obtained the topographic image using contact mode, from which the sample surface tilting along the fast and slow scan axes could be compensated. Then the tip lifted the sample surface, and MFM images were obtained in frequency modulation mode. The dither of the piezo plate causes the cantilever to oscillate. The tip holder was hooked with a spring by a beryllium copper. The spring tension drove the probe holder mostly to the microscope’s bottom. The tip holder can only move in a small area when subjected to a specific external force, which increases the ability to locate small samples in the microscope [19]. In general, a piezo-resistive cantilever is excited to oscillate at its resonant frequency by such a positive feedback loop (W_0_). At the cantilever’s free end is a magnetic coated tip. The force gradient change ΔF (X, Y, Z) that occurred when we scanned the tip over a sample surface might cause a shift within the cantilever’s efficient resonant frequency (ΔW_0_).

## 3. System Design

### 3.1. Cooling System

These examinations have been achieved by the Oxford Shanghai demonstrative facility using a cryogen-free superconducting magnetic device (Teleprompt by Oxford Devices, which provides top loading access to a sample in a variable magnetic field / low-temperature environment, seen in Figure 2a. The magnetic field produced through the system applied in this experiment could achieve 12 T. The magnet is a 12 T magnet and because we want to make sure it is running safely, we normally run it at 11 T maximum. An integrated VTI having sample temperatures ranging between 1.6 to 300 K was included, as well as an enclosed cylindrical sample space with a diameter of 50 mm. When the MFM head is introduced into the sample space, both the sample and pre-amplifier road are in an ambient helium gas and are cooled mainly through a stable exchange of helium gas [20,21]. The 12 T superconducting magnets are housed just at the cryostat’s bottom. When the pre-amplifier road is fitted into the VTI, the samples remain inside the magnet’s core [22].

### 3.2. Design of Pre-Amplifier Board

The following Figure 2b shows the composition of a pre-amplifier circuit box with 75 mm as the outer diameter and a height of 39 mm; a bellow joint with 25 mm as the outer diameter; a core tube with a maximum diameter of 10 mm; seven radiation shield plates with an approximate diameter of 46 mm; a heavy rod with an external diameter of 35 mm made of Teflon as well as brass; and a sapphire interface piece with multiple pins. The circuit box is mounted at the top end of the central tube and therefore is connected to it by the bellows joint, housing the pre-amplifier circuit. The circuit box is the only component that is not inside the magnet. The pre-amplifier board, the inner component, which is enclosed in the magnet, is made up of the remaining sections. Soft bellow joints can protect the preamplifier board inside from the cryogen-free magnet system’s vibration and the bottom of the preamplifier circuit box is a standard KF50 flange, which can vacuum seal with the flange provided at the entrance of the magnet. Stainless steel 304 was used to make this three-radiation shielding plate. Effective heat exchange among the MFM heads under low temperature and the external at ambient temperature is decreased by polishing both sides of each plate. They are mounted on the center tube in a coaxial configuration. A bottom end of a central tube such as this is fixed to a heavy rod, and the main tube is concentrically connected toward the heavy rod throughout the series, with such a long through hole in the center allowing the flowing wire. The heavy rod was supposed to lower the resonance frequency of the pre-amplifier road, hence improving vibration isolation. This heavy rod puts three Teflon and three brass weights in series. This configuration can help reduce heat exchange between the device’s top and bottom ends. Since this heavy rod is thick and long, it is likely to generate a severe thermal short in the cold bore of the magnet, which is usually filled with low-pressure helium gas for temperature exchange. This sapphire interface piece was attached directly to the heavy rod underneath. The vibration-isolating spring holds the MFM head underneath the heavy rod. The heavy rod can help ensure that the MFM head is hanging in the middle of the magnet bore due to gravity. The measurement and control output wires from the pre-amplifier board go upwards while connected to the sapphire interface piece at the bottom from the top side of the sapphire interface piece, new wires go upward, pass through the central hole of the heavy rod and central tube, and are connected to the vacuum sealed interface (made of sapphire) at the bottom of the pre-amplifier circuit box.

### 3.3. MFM Head

The MFM head hangs on a spring from the bottom of the pre-amplifier board, as illustrated in Figure 2b. As described in Section 1, achieving a small size is the major challenge for building a high-resolution MFM at a low temperature. To solve this problem, we designed the MFM head based on the piezoelectric tube (PZT), as seen in Figure 3. We utilized a SpiderDrive, generally called a piezoelectric motor, in this MFM [16]. This motor is extremely rigid and compact, so it performs excellently in demanding environments, much like the Chinese Academy of Sciences’ High Magnetic Field Laboratory’s hybrid magnet [23]. The motor consists mainly of the piezoelectric tube (PZT), zirconia tube, tantalum square shaft, and a beryllium copper spring plate. PZT (four-quadrant, thickness 0.55 mm, EBL#3 type material, from EBL. Products Inc. 6802CE-2RS from Xundazc Company), with a diameter of 7.09 mm in 42 mm. The guiding tube was 18 mm long and made of oxide zirconia. The inner part was properly polished, and the outer diameter was slightly longer with one end so that it could match and subsequently be glued with the inner side of the PZT’s free end (the other end of PZT is known as the mounting end, which will be affixed to the bottom of an MFM structure).

The sliding shaft and the spring plate were made of high-purity tantalum and beryllium copper, respectively. The sliding shaft is 28 mm in length and a square cross-section that fits perfectly into a guiding tube using a bent spring plate is placed into the gap between both the sliding shaft’s outer flat surface and the internal part of the guiding tube, allowing a shaft to be spring-loaded and driven by the PZT in inertial mode. The principles for the operation of SpiderDrive can be found in the related literature [23]. The SpiderDrive have being continuously compacted and has significance for such applications. A sample was fixed on the higher side of the shaft, and the cantilever holding was affixed under a crossbeam under 12 T Cryogen of Free Superconducting Magnet Systems.

## 4. Performance Test

The imaging accomplishment of the MFM head structure in the cryogen-free magnet is shown in Figure 4 and Figure 5, which display the optical image of the cantilever tip and the lateral alignment of the tip and sample. Before measurement, we moved the whole device under the optical microscope and detect the relative position of the sample to the tip through the window of the MFM holder. Gradually, we adjusted the position between the two by affecting the MFM holder with a pair of tweezers. The tip is aligned with the sample area. The tip position above the sample can be controlled by an optical microscope, for a located small sample area, which is shown in Figure 4a,b at room temperature. The scanning area at room temperature is 50 µm and at low temperature is 10 µm; the scanning tube is not only for scanning but also adjusts for (Z) distance if the tip and sample are large in the tube to the left of the tip to the attractive position. The resonant frequencies of the scanner are around 1000 Hz. In the range of temperatures at which studies can be carried out, the VTI of low temperatures is 1.6 K but the cantilever tip will heat the scanning area, so the real temperature is 5 K to 300 K. For the topography scanning, we used the tapping mode, and for the MFM imaging, we used the non-contact mode. The scanning speed for topography is very slow (4 s per line) and for MFM imaging it is (1.7 s per line). The resonant frequency of the cantilever is (33.59 kHz) and the quality factor of the cantilever is (1000).

Figure 5a–h shows the magnetic image of the topography and videotape tracks. The best and most effective strategy to confirm that the MFM system is operating is to capture the track layout on a videotape. A unique device was used to study and scan in succession at 5 K, and we applied a magnetic field vertical to the sample and gradually increased the magnetic field from 0 to 11 T. Videotape samples, there have magnetic track polders with tiny patterns connected areas. When we applied the magnetic field, some areas became fully saturated, indicating that each particle is the magnetization spin of an aligned particle, meaning that it was completely saturated. However, when we performed the scan, the tip still had magnetic tracks because this area was without magnetic particles (dark area) and other areas had magnetic particles. The image contrast appears smaller because of a high magnetic field. The magnetic track polders were saturated by less than 11 T at the earlier stage [24]. When the magnetic field increases, the magnetic polders are saturated, so it is necessary to prevent the tip-sample distance from getting larger and larger because the external magnetic field applies a much bigger force on the cantilever; if we do not withdraw the tip, the cantilever will bend and crush the tip on the sample surface. Nevertheless, we still observed weak patterns because the tip-sample distance is larger, as is shown in Figure 5f. The MFM head system under the cryogen magnet has several practical applications.

## 5. Conclusions

In summary, the present work is significant in that we built a new design of the MFM head that can work stably in the Cryogen Free Superconducting Magnet, which can be achieved with a maximum magnetic field strength of 11 T. The results confirm that this is a good choice for features of new structures and that the construction of the novelty MFM head demonstrated great compactness and rigidity. Our results provide several feasible and extensive potential applications in heterostructure materials and some small device applications in the future.

## Figures and Tables

**Figure 1 micromachines-13-01922-f001:**
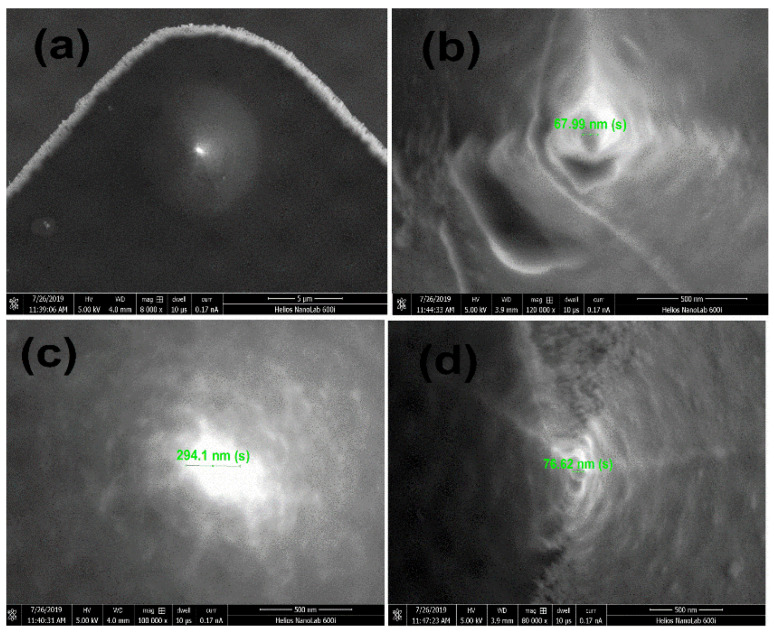
SEM image of a piezoresistive cantilever of magnetic film at the coated tip with different magnifications (**a**) 100 nm (**b**) 200 nm (**c**) 300 nm and (**d**) 500 nm.

**Figure 2 micromachines-13-01922-f002:**
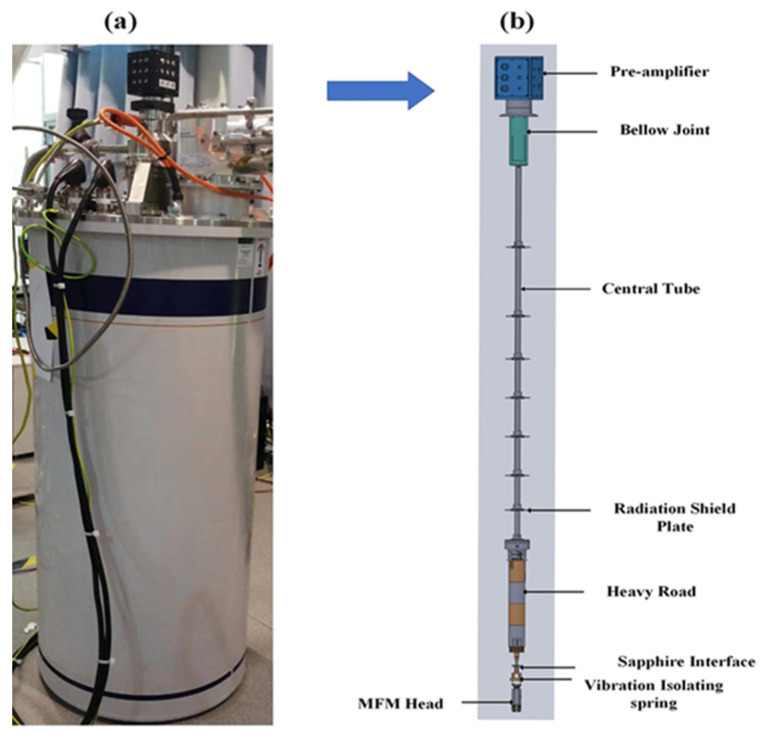
(**a**) Photograph and cross-section of cryogen-free superconducting magnet system and (**b**) photograph of the pre-amplifier board.

**Figure 3 micromachines-13-01922-f003:**
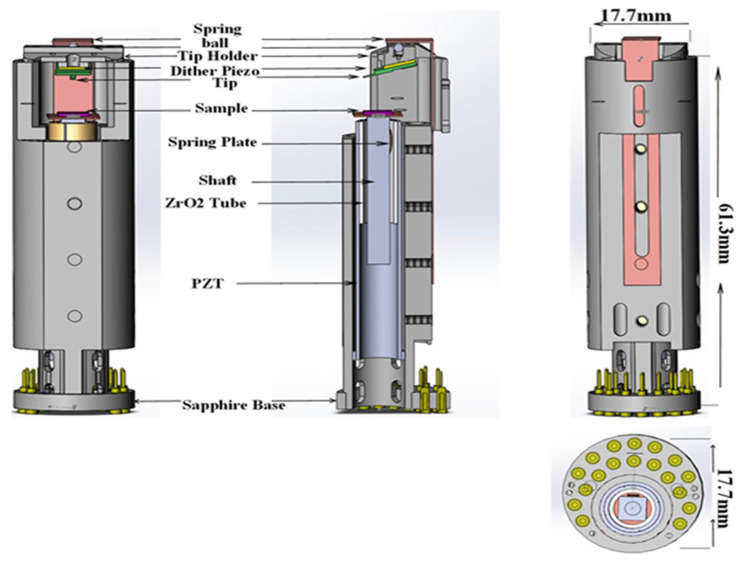
MFM head view (left) with side image (middle). The MFM’s internal side view (top image) and bottom side (lower image) are shown in the images on the right.

**Figure 4 micromachines-13-01922-f004:**
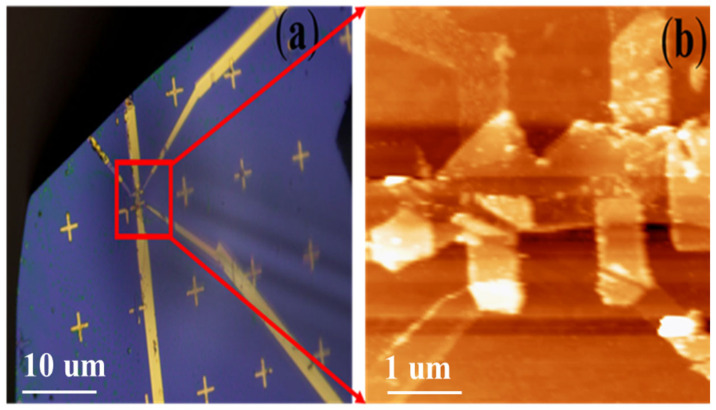
(**a**) An image of the sample under an optical microscope with a scale of 10 µm. (**b**) MFM measurement topography by the scale bar is 1 µm, measured at room temperature.

**Figure 5 micromachines-13-01922-f005:**
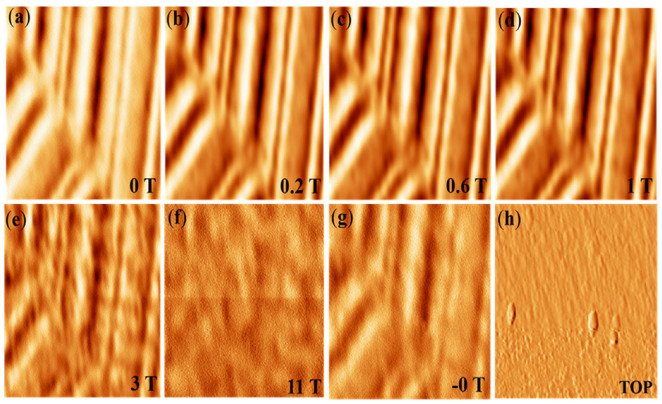
(**a**–**g**): Evaluation domain of commercial videotape tracks at 5 K with changing magnetic field from 0 T to 11 T; the tip-sample distance (**a**) is 100 nm and (**f**) is 500 nm, respectively; the image size is 12 µm × 12 µm, and (**h**) shows the topography image of the same scan area.

## Data Availability

The data that support the findings of this study are availablefrom the corresponding author upon reasonable request.

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
