# Peer review of "Compact Magnetic Force Microscope (MFM) System in a 12 T Cryogen-Free Superconducting Magnet"

_micromachines, 2022, doi:10.3390/mi13111922_

Round 1
Reviewer 1 Report
The authors have developed a compact AFM for operation at low temperatures and strong magnetic fields. Original engineering solutions were used, which could be of interest to the reader, however, in the presented form, the manuscript is difficult to recommend for publication.
1. English language. The article should be rewritten, paying attention to the quality of the translation into English.
2. Terminology. In scanning probe microscopy, the term "probe" is used for the part that comes into contact with the surface. In the case of AFM, this is usually a cantilever. Using this term to refer to other parts of the microscope or to the microscope as a whole (as done in section 3.2) leads to misunderstandings.
3. Performance test. Instrument test results should be explained or rechecked. It is not clear why the magnetic structure is clearly visible in fields of 1 and 3 T and disappears only in a field of 11 T. Typically, the videotape coercivity field is less than 0.3T. Even more surprising is that after applying the 11 T field and returning to the 0 T field, the original magnetic structure appears.
4. The technical characteristics of the AFM should be indicated: the scanning area, the resonant frequencies of the scanner, the range of temperatures at which studies can be carried out. Specify topography scanning mode and the mode of MFM imaging, scanning speed, frequency and quality factor of the cantilever.
5. The article can be shortened. Part of the introduction from line 58 (The system's dimensionality has been one...) to line 78 is not related to the topic of the article.
Minor points:
line 137: amplifier circuit box with 75cm as outer diameter... May be 75 mm?
line 201: MFM measurement topography by the scale bare is 1 um, at... a) There is no a scale bar. b) Typically MFM refers to magnetic imaging and AFM refers to topography.
line 213: The unique device was used to study and scan in succession at 1.6 K, and then we applied a magnetic field vertical to the sample and gradually increased the magnetic field from 0 T to 11 T parallel to the sample surface;... Is the field vertical or parallel?
Author Response
Editor #:
Please check that all references are relevant to the contents of the manuscript.
Response: Thank you for pointing out this comment. We have revised the manuscript accordingly; we have checked all the references in the revised manuscript.
Reviewer #1:
- English language. The article should be rewritten, paying attention to the quality of the translation into English.
Response: Thank you for the mention. We have revised the manuscript accordingly; we have read and checked the manuscript on our own and with a native speaker. We made some changes in the revised manuscript as marked in red.
2. Terminology. In scanning probe microscopy, the term "probe" is used for the part that comes into contact with the surface. In the case of AFM, this is usually a cantilever. Using this term to refer to other parts of the microscope or to the microscope as a whole (as done in section 3.2) leads to misunderstandings.
Response: Thank you for pointing out this comment. We have revised the manuscript accordingly; we have corrected, the term (probe) in the revised manuscript.
3. Performance test. Instrument test results should be explained or rechecked. It is not clear why the magnetic structure is clearly visible in fields of 1 and 3 T and disappears only in a field of 11 T. Typically, the videotape coercivity field is less than 0.3T. Even more surprising is that after applying the 11 T field and returning to the 0 T field, the original magnetic structure appears.
Response: Thank you for figuring out this comment. We have revised the manuscript accordingly; in the low field the tracks magnetized are vertical, and we can still see the image contrast but when the field goes very high, there is a force between tip and sample; to handle that, we need to lift tip going higher and higher, so the profile of force gradient become pretty small, that is why in the very high field still see the contrast, this because the tip is far away from the sample surface, so the contrast is smaller, but it is never saturated. The saturated means, the contrast is uniform. The resulting image doesn’t show uniformity, even if we move the tip very high from the surface still has profile contrast. The profile contrast is getting smaller and smaller. In a low field, everything will be saturated and the coercivity is pretty low as known the field is higher than the coercivity of everything saturated, even if saturated we still see the contrast, because there are polders and why the contrast gets smaller because the force between tip and sample is a function of the magnetic field. When we increase the field, we need to withdraw the tip to higher, then the profile of the tip senses smaller. Smaller means doesn’t disappear is never disappears even in high fields. We have added it to the revised manuscript.
4. The technical characteristics of the AFM should be indicated: the scanning area, the resonant frequencies of the scanner, the range of temperatures at which studies can be carried out. Specify topography scanning mode and the mode of MFM imaging, scanning speed, frequency and quality factor of the cantilever.
Response: Thank you for pointing out this comment. We have revised the manuscript accordingly; the scanning area at room temperature is 50µm and at low temperature is 10µm, the scanning tube is not only for scanning but also adjusts for (Z) distance, if the tip and sample are large in the tube to the left the tip to the attractive position. The resonant frequencies of the scanner are around 1000Hz. In the range of temperatures at which studies can be carried out, the VTI of low temperatures is 1.6 K but the cantilever tip will heat the scanning area, so the real temperature is 5 K to 300 K. For the specific topography scanning mode, we used by tapping mode, while the MFM imaging, we used non-contact mod. The scanning speed for topography is very slow (4 seconds per line) and for MFM imaging is (1.7 seconds per line). The resonant frequency of the cantilever is about (33.59 kHz) and the quality factor of the cantilever is (1000). We have added it to the revised manuscript.
5. The article can be shortened. Part of the introduction from line 58 (The system's dimensionality has been one...) to line 78 is not related to the topic of the article.
Response: Thank you for your good suggestion pointing. We have revised the manuscript accordingly; the part of the introduction has been deleted from the revised manuscript.
Minor points:
line 137: amplifier circuit box with 75cm as outer diameter... May be 75 mm?
Response: Thank you for this comment. We have revised the manuscript accordingly; the amplifier circuit box has been corrected in the revised manuscript.
line 201: MFM measurement topography by the scale bare is 1 um, at... a) There is no a scale bar. b) Typically MFM refers to magnetic imaging and AFM refers to topography.
Response: Thank you for this comment. We have revised the manuscript accordingly; we have modified the scale bar in the revised manuscript.
line 213: The unique device was used to study and scan in succession at 1.6 K, and then we applied a magnetic field vertical to the sample and gradually increased the magnetic field from 0 T to 11 T parallel to the sample surface;... Is the field vertical or parallel?
Response: Thank you for this comment. We have revised the manuscript accordingly; the magnetic field is always vertical, to avoid this confusion, we have deleted this sentence from the revised manuscript.

Reviewer 2 Report
Dear Authors,
Please find my comments regarding the paper A compact new probe-type Magnetic Force Microscope (MFM) for use in 12 T Cryogen-Free Superconducting Magnet by Asim Abas, Tao Geng, Wenjie Meng, Jihao Wang, Qiyuan Feng, Hou Yubin and Qingyou Lu.
In line 133 you mention 11 T magnet used for experiments while in rest you mention 12 T. Please change accordingly.
Again, figure 3 presents magnet up to 11 T not 12 T. Should we believe that the magnet is of 12 T but you only worked at 11 T? Please clarify this aspect because in the conclusion you mention again 12 T.
Otherwise the paper is quite extensive described and the work made here is very relevant for further studies.
One more aspect probably will increase the visibility and potential of the work, if the authors presents the coating of the tip. Some SEM images, to measure (maybe via FIB SEM) the thickness of the coating. The authors mention that the “The cantilever beam's tip has to be covered with a magnetic coating in order to detect magnetic force” (This statement raises question…. what coating? What material? what thickness?) and then 50 nm of cobalt and 5 nm of gold. How did the authors measure this thickness? And again, for the magnetic coating how the thickness of the coating affects the produced MFM signal? Probably, maybe not necessarily in this work (this is the design and proof of concept), the authors should extend their research modifying the thickness of magnetic coating coating….
Nevertheless, the paper can be accepted for publication as is, after clarifying the power of the magnet used for the experiments.
Author Response
Editor #:
Please check that all references are relevant to the contents of the manuscript.
Response: Thank you for pointing out this comment. We have revised the manuscript accordingly; we have checked all the references in the revised manuscript.
Reviewer #2:
In line 133 you mention 11 T magnet used for experiments while in rest you mention 12 T. Please change accordingly.
Response: Thank you for your advice. We have revised the manuscript accordingly; the magnet is a 12 T magnet and because we want to make sure is running safely, so we normally run out in 11T maximum. We have added it to the revised manuscript.
Again, figure 3 presents magnet up to 11 T not 12 T. Should we believe that the magnet is of 12 T but you only worked at 11 T? Please clarify this aspect because in the conclusion you mention again 12 T.
Response: Thank you for your consideration. We have revised the manuscript accordingly; the magnet is a 12 T magnet and because we want to make sure is running safely, so we normally run out in 11T maximum. We have added it to the revised manuscript.
Otherwise the paper is quite extensive described and the work made here is very relevant for further studies.
One more aspect probably will increase the visibility and potential of the work, if the authors presents the coating of the tip. Some SEM images, to measure (maybe via FIB SEM) the thickness of the coating. The authors mention that the “The cantilever beam's tip has to be covered with a magnetic coating in order to detect magnetic force” (This statement raises question…. what coating? What material? what thickness?) and then 50 nm of cobalt and 5 nm of gold. How did the authors measure this thickness? And again, for the magnetic coating how the thickness of the coating affects the produced MFM signal? Probably, maybe not necessarily in this work (this is the design and proof of concept), the authors should extend their research modifying the thickness of magnetic coating coating….
Response: We sincerely thank the reviewer for the suggestion. We have revised the manuscript accordingly; we have added all this information and also, and we put the SEM image into the revised manuscript.

Round 2
Reviewer 1 Report
The authors have improved the article, but one problem remains.
Explanation (lines 233-246) of Fig. 5 is unclear and should be rewritten in better English. As far as I understand, several statements can be distinguished from it, which, in turn, require clarification.
1. The videotape saturation field at 5K exceeds 11 T. (Is there any confirmation of this notice?)
2. 2. As the magnetic field increases, the contrast becomes lower due to problems with measurements in strong fields. (What is the real range of MFM measurement fields? Why does the force between the probe and the sample depend on the external magnetic field?)
3. 3. The initial magnetic pattern still exists at 11 T, but we can't see it because for some reason the probe was raised high above the surface. (Why is it necessary to increase the distance?)
Author Response
Explanation (lines 233-246) of Fig. 5 is unclear and should be rewritten in better English. As far as I understand, several statements can be distinguished from it, which, in turn, require clarification.
Response: Thank you for your comment. We have revised the manuscript accordingly; we have read and checked the manuscript on our own and with a native speaker. We made some changes in the revised manuscript as marked in red.
The videotape saturation field at 5K exceeds 11 T. (Is there any confirmation of this notice?)
Response: Thank you for pointing out this comment. We have revised the manuscript accordingly; On the videotape sample, there are magnetic track polders with teeny patterns in the district on connected areas. Then when we apply the magnetic field, these areas are fully saturated, meaning that each particle is the magnetization spin of a particle aligned up, it means completely saturated. However, when we scan, the tip still has magnetic tracks because this area is without magnetic particles (dark area) and another area with magnetic particles. We have added this information to the revised manuscript.
As the magnetic field increases, the contrast becomes lower due to problems with measurements in strong fields. (What is the real range of MFM measurement fields? Why does the force between the probe and the sample depend on the external magnetic field?)
Response: Thank you for figuring out this comment. We have revised the manuscript accordingly; the real range of the MFM measurement field is from 0 to 11 T. Secondly When the magnetic field goes up is increasing, the magnetic polders are saturated already, so we need to withdraw the tip-sample distance getting larger, and larger because the external magnetic field applied a much bigger force on the cantilever; if we don't withdraw the tip, the cantilever caused to bend and crash the tip on the sample surface. We have added this information to the revised manuscript.
The initial magnetic pattern still exists at 11 T, but we can't see it because for some reason the probe was raised high above the surface. (Why is it necessary to increase the distance?)
Response: Thank you for the mention. We have revised the manuscript accordingly; The image contrast is seen as smaller because of a high magnetic field. The magnetic track polders were saturated by less than 11 T at the earlier stage, so we need to withdraw the tip-sample distance getting larger and larger because the external magnetic field applied a much bigger force on the cantilever; if we don't withdraw the tip, the cantilever caused to bend and crash the tip on the sample surface. Nevertheless, we still see weak patterns because the tip-sample distance is larger[1]. We have added this information and this reference to the revised manuscript.
Ref
- Xiang, K.; Hou, Y.; Wang, J.; Zhang, J.; Feng, Q.; Wang, Z.; Meng, W.; Lu, Q.; Lu, Y. A Piezoelectric Rotatable Magnetic Force Microscope System in a 10 T Cryogen-Free Superconducting Magnet. Rev. Sci. Instrum. 2022, 93, 093706, doi:10.1063/5.0100662.
